# Direct Injection of Recombinant AAV-Containing Solution into the Oviductal Lumen of Pregnant Mice Caused In Situ Infection of Both Preimplantation Embryos and Oviductal Epithelium

**DOI:** 10.3390/ijms23094897

**Published:** 2022-04-28

**Authors:** Masahiro Sato, Nami Sato-Yamamoto, Ai Wakita, Misako Haraguchi, Manabu Shimonishi, Hiroyuki Okuno

**Affiliations:** 1Section of Gene Expression Regulation, Frontier Science Research Center, Kagoshima University, Kagoshima 890-8544, Japan; 2Graduate School of Medical and Dental Sciences, Kagoshima University, Kagoshima 890-8544, Japan; sato_nami@kacnet.co.jp (N.S.-Y.); k4169763@kadai.jp (A.W.); haraguci@m3.kufm.kagoshima-u.ac.jp (M.H.); 3Biological Science Center, KAC Co., Ltd., Shiga 520-3001, Japan; shimonishi_manabu@kacnet.co.jp

**Keywords:** adeno-associated virus, serotype 6, preimplantation embryo, intra-oviductal injection, enhanced green fluorescent protein, GONAD, morula, mouse, transduction, oviductal epithelium, zona pellucida-enclosed embryo, ex vivo handling of embryo

## Abstract

Adeno-associated virus (AAV) vector is an efficient viral-based gene delivery tool used with many types of cells and tissues, including neuronal cells and muscles. AAV serotype 6 (AAV-6), one of numerous AAV serotypes, was recently found to efficiently transduce mouse preimplantation embryos. Furthermore, through coupling with a clustered, regularly interspaced, short palindromic repeat (CRISPR)/CRISPR-associated protein 9 (Cas9) system—a modern genome editing technology—AAV-6 has been shown to effectively create a mutation at a target locus, which relies on isolation of zygotes, in vitro viral infection, and transplantation of the infected embryos to recipient females. Unfortunately, this procedure, termed “ex vivo handling of embryos”, requires considerable investment of capital, time, and effort. Direct transduction of preimplantation embryos through the introduction of AAV-6 into the oviductal lumen of pregnant females would be an ideal approach. In this study, we injected various types of recombinant AAV vectors (namely, rAAV-CAG-EGFP-1, -2, -5, and -6, each carrying an enhanced green fluorescent protein [*EGFP*] cDNA whose expression is under the influence of a cytomegalovirus enhancer + chicken β-actin promoter) into the ampulla region of oviducts in pregnant female mice at Day 0.7 of pregnancy (corresponding to the late 1-cell stage), and EGFP-derived green fluorescence was assessed in the respective morulae. The highest levels of fluorescence were observed in rAAV-CAG-EGFP-6. The oviductal epithelium was distinctly fluorescent. The fluorescence in embryos peaked at the morula stage. Our results indicate that intra-oviductal injection of AAV-6 vectors is the most effective method for transducing zona pellucida-enclosed preimplantation embryos in situ. AAV-6 vectors could be a useful tool in the genetic manipulation of early embryos, as well as oviductal epithelial cells.

## 1. Introduction

Clustered, regularly interspaced, short palindrome repeats (CRISPR)/CRISPR-associated protein (Cas9) is an effective and simple genome editing platform (reviewed by Harrison et al. [1] and Hsu et al. [2]). Here, a purified Cas9 endonuclease protein is complexed with a single guide RNA (sgRNA, i.e., ribonucleoprotein [RNP]), and is introduced into a cell where the complex binds to a target locus and causes a double-stranded break (DSB) in the absence of an oligodeoxynucleotide or donor sequence (one showing homology to the target sequence). Then, the cleaved site generates insertion or deletion of sequences during the process of DNA repair. In the presence of the donor sequence, homology-directed repair (HDR) and indels occur at the target site. Notably, DSB is known to occur in both dividing and non-dividing cells, but HDR preferentially occurs in dividing cells [3].

Adeno-associated viruses (AAVs) are non-pathogenic, non-enveloped, single-stranded DNA viruses that allow highly efficient gene delivery to a broad range of host cells (both dividing and non-dividing cells), and elicit a low immune response [4]. The viral particle is approximately 20 nm in diameter and can carry a genomic load of up to 4.5 kilobase (kb). Recombinant AAV (rAAV) vectors can be produced by replacing the wild-type (WT) coding regions with any gene or DNA sequence of interest. Upon infection of target cells, the viral genome must first be converted into a transcriptionally active double-stranded form, which is a crucial rate-limiting step in transgene expression using an rAAV vector (reviewed by Ferrari et al. [5]). The strategy considers self-complementary AAV (scAAV) vectors that can bypass the rate-limiting step for second-strand synthesis [6]. Upon transduction of target cells, the scAAV genome exists either as circular genomes or concatemers, with the former considered to be much more effective in transgene expression [7].

By manipulating target genome sequences, through simple co-incubation with zona pellucida (ZP)-enclosed preimplantation mammalian embryos, rAAV vectors can be used to produce genetically modified (GM) animals, such as mice [8,9,10,11,12,13], rats [8], bovines [8], and a nonhuman primate (cynomolgus monkey) [14]. Given the large number of AAV serotypes (at least 14), researchers must validate the performance of an AAV serotype in the transduction of preimplantation mammalian embryos. Yoon et al. [9] first explored the possibility of using ZP-intact mouse morulae through co-cultivation with AAV vectors. Of the 14 rAAV vectors (carrying an enhanced green fluorescent protein [*EGFP*] transgene), all were found to successfully transduce morulae. Serotype 6 rAAV (rAAV-6) exhibited a particularly high degree of fluorescence. Mizuno et al. [8] obtained similar results. Lee et al. [10] concluded that rAAV-1 and rAAV-6 provided better transduction efficiencies in early mouse embryos. Romeo et al. [13] demonstrated that rAAV-1 or rAAV-DJ diffused efficiently across the ZP to deliver genes into zygotes. Interestingly, both ZP-intact rat and bovine embryos were effectively infected by rAAV-6 [8]. These findings suggest that the rAAV-mediated approach offers a simple and efficient method to manipulate the genomes of a broad range of mammalian species.

Mizuno et al. [8] and Yoon et al. [9] first assessed the ability of rAAV-6 to induce CRISPR-based genome editing in ZP-intact zygotes. Yoon et al. [9] demonstrated that when zygotes were incubated in the presence of rAAV6-*Cas9* (carrying *Cas9* gene under the control of a mouse *U1a* small nuclear RNA promoter) and rAAV6-g*Tyr* (carrying an *EGFP* expression unit together with a guide RNA [gRNA] expression unit targeting the tyrosinase [*Tyr*] gene, which encodes for a protein involved in melanin synthesis) for 3 d (up to the blastocyst stage), the blastocysts exhibited indels at *Tyr* with 100% efficiency. This clearly demonstrates the efficacy of the AAV capsid-based delivery of genetic components inside the embryo that allows for genome editing at the target locus.

The available methods for generating GM animals require ex vivo handling of embryos, which includes “isolation of zygotes from female oviducts”, “in vitro delivery of genome editing components” (via microinjection [MI], electroporation [EP], or simple co-incubation with rAAV), “cultivation of the genome-edited zygotes prior to egg transfer (ET)”, and “ET of genome-edited embryos to the reproductive tracts (oviduct or uterus) of a pseudo-pregnant recipient female”. These processes require considerable amounts of capital, time, and labor. Takahashi et al. [15] successfully bypassed these laborious ex vivo steps by performing in situ genome editing of preimplantation embryos (2-cell embryos) floating in the oviductal lumen. This was done by direct injection of a solution (1–1.5 μL) containing *Cas9* mRNA and sgRNA into the ampullar lumen, an expanded area of the oviduct, under a dissecting microscope. After instillation, the entire oviduct was subjected to in vivo EP using tweezer-type electrodes. When developing mid-gestational fetuses were analyzed, almost all exhibited mutations at the target locus. This technology has been termed “Genome-editing via Oviductal Nucleic Acid Delivery (GONAD)”. The modified version of GONAD, “*improved* GONAD (*i*-GONAD)”, was developed for genome editing in zygotes at Day 0.7 of gestation (corresponding to late 1-cell stage) after introduction of RNP (with a donor sequence in some cases) [16].

Yoon et al. [9] demonstrated that infection of ZP-intact preimplantation embryos floating in the oviductal lumen is possible by a simple intra-oviductal instillation of a rAAV-containing solution, similar to GONAD. Briefly, they injected a solution containing the rAAV6-*Cas9* and rAAV6-g*Tyr* vectors into the oviduct of female mice at Day 0.5 of pregnancy. Of the 29 pups birthed, three were found to have indels. The mutated founder mice generated albino offspring, which indicated germline transmission. This suggests that AAVs can deliver CRISPR/Cas9 components to ZP-enclosed early embryos in vivo. This approach has been termed “AAV-based GONAD” (reviewed by Sato et al. [17]). To date, the procedure of AAV-based GONAD has been largely neglected in scientific publications, regardless of its potential in laboratory applications. In the present study, we asked the following research questions: (1) which types of rAAVs are effective for in vivo transduction of preimplantation mouse embryos? (2) which stages of preimplantation embryos show the best performance in transgene expression after AAV-based GONAD? and (3) is the oviductal epithelium (facing oviductal lumen) infected after AAV-based GONAD? We have performed some critical evaluation using rAAVs carrying an *EGFP* expression unit.

## 2. Results

We first confirmed the transducing ability of our purified rAAVs in HEK293T cells. All serotypes (scAAV-CAG-EGFP-1, -2, -5, and -6) were used at final concentrations of 1–2 × 10^9^ viral genome copies (GC)/mL in the culture medium. When cells were inspected for EGFP-derived fluorescence 3 days after transduction, most cells exhibited bright fluorescence, irrespective of the serotype (Appendix A), suggesting that all rAAVs were active.

### 2.1. Experiment 1: AVV-Based GONAD Using Serotype 6 of rAAV

In previous studies, rAAV-6 was particularly useful in transducing preimplantation mouse embryos (zygotes to morulae) [8,9,10]. We injected 1 μL of a solution containing scAAV-CAG-EGFP-6 (3 × 10^12^ GC/mL) and 0.4% FastGreen FC into the oviductal lumen upstream of the ampulla at Day 0.7 of pregnancy (Figure 1A). Two days after AAV-based GONAD, oviducts were dissected to isolate the morulae. The isolated morulae were assessed for EGFP-derived green fluorescence. In Figure 1B(a–f), fluorescence images for morulae isolated from one pregnant female treated with AVV-based GONAD (Figure 1B(a–c)), and those isolated from one untreated control pregnant female (Figure 1B(d–f)), are shown. Notably, in the former group, embryos with varying intensity were discernible (Figure 1B(b,c)). Optical sectioning images further revealed the specificity of EGFP-derived fluorescence (Appendix A). In contrast, only weak autofluorescence was observed in the latter group (control embryos) (Figure 1B(d–f)).

In our previous experiments, we demonstrated that the exogenous substance (i.e., plasmid DNA and fluorescent dextran) can be incorporated by the oviductal epithelial cells facing the oviductal lumen when entire oviducts were subjected to in vivo EP after oviductal instillation [18]. To examine the possible transduction of the oviductal epithelial cells with scAAV-CAG-EGFP-6, we assessed the expression of fluorescence in the oviducts 2 d after AAV-based GONAD. As expected, bright fluorescence was discernible at the injection site (arrow in Figure 1B(h,i)). No GFP fluorescence was noted in the control oviducts (Figure 1B(j–l)). The results of our two-day in situ experiment indicated that both preimplantation mouse embryos and oviductal epithelial cells can be successfully transduced with the serotype 6 of rAAV vector.

### 2.2. Experiment 2: AVV-Based GONAD Using Various Serotypes of rAAV

The other serotypes of rAAV (except rAAV-1) are largely inefficient in the transduction of preimplantation mouse embryos in vitro [8,9,10]. To validate these findings, we performed AAV-based GONAD using scAAV-CAG-EGFP-1, -2, -5 and -6. The virus titers were similar to those in Experiment 1 (2–3 × 10^12^ GC/mL). In the morulae, embryos transduced with scAAV-CAG-EGFP-6 exhibited the strongest fluorescence, those with scAAV-CAG-EGFP-1 exhibited moderate fluorescence, and those with scAAV-CAG-EGFP-2 and -5 showed little or no fluorescence (Figure 2A). We assessed the fluorescence intensity of each embryo according to regions of interest (ROIs). The GFP intensity in the embryos transduced with scAAV-CAG-EGFP-6 was significantly higher than that in the control embryos (*p* < 0.005, Figure 2B). In contrast, no statistically significant difference was noted among the embryos transduced with scAAV-CAG-EGFP-1, -2 and -5 or the control embryos, though a few bright GFP signals were observed in some scAAV-CAG-EGFP-1-infected embryos (arrows in Figure 2A(b,c),B).

After infection with scAAV-CAG-EGFP-1 and -6, the oviductal epithelial cells exhibited strong fluorescence at the injection site (arrows in Figure 2A(d,p)), while scAAV-CAG-EGFP-2 and -5 elicited little, or no, fluorescence (arrowheads in Figure 2A(h,l)). Notably, we observed a modest, but significant, positive correlation between transduction performance in the embryos and that in the oviductal epithelial cells (*r* = 0.5217, *p* < 0.0001, Figure 2C).

### 2.3. Experiment 3: Expression Analysis of EGFP in Zygotes after AVV-Based GONAD

We examined the duration of EGFP expression during the 2-cell to late blastocyst stage, when AAV-based GONAD was carried out at the late zygote stage using scAAV-CAG-EGFP-6 (6 × 10^12^ GC/mL, Figure 3A). One day after virus instillation, two-cell embryos were isolated and subjected to in vitro cultivation until the late blastocyst stage. During cultivation, fluorescence was inspected daily at the 2-cell (Day 1.5), morula (including 8-cell embryos, Day 2.5), early blastocyst (Day 3.5), and late blastocyst stages (Day 4.5). Fluorescence was discernible at the 2-cell stage and peaked at the morula stage (Figure 3B). Fluorescence decreased during development from morulae to late blastocysts (Figure 3B). Repeated measures of one-way analysis of variance (ANOVA) of GFP fluorescent signals in individual embryos revealed a significant change in fluorescence (*p* < 0.0001) and a peak at the morula stage (*p* < 0.0003, Figure 3C,D), even though there is a high variability of fluorescent levels in individual embryos.

## 3. Discussion

From the zygote to early blastocyst stage, a preimplantation mouse embryo is surrounded by a ZP, a translucent multilayered porous matrix of glycoproteins that protects the embryo against harmful substances, including viruses (reviewed by Wassarman [19,20]). The ZP is one of the rate-limiting factors influencing the efficiency of gene delivery. Mammalian preimplantation embryos have been particularly difficult to transfect with traditional methods used in somatic cells. The most commonly used method for gene delivery through the ZP is the physical pronuclear MI of purified DNA [21]. To accelerate plasmid DNA delivery in ZP-enclosed embryos, in vitro EP has often been employed. However, the optimal electrical conditions have to be carefully considered [22]. Furthermore, ZP must be weakened by a brief acidic treatment of Tyrode’s solution before EP, which may enhance the uptake of exogenous DNA and protect the embryos from electroporation damage [23,24,25]. Virus-mediated approaches, such as retrovirus (RV), adenovirus (AD), and AAV vectors, have also been developed for the transduction of preimplantation embryos. AD and AAV vectors can transduce preimplantation embryos located in the perivitelline space (the space between the ZP and the membrane of the oocyte) through MI of viral elements, laser perforation of the ZP [26], or a medium containing ZP-free embryos [27,28]. AAV vectors can infect ZP-enclosed embryos, although the infection ability is dependent on the serotype [8,9,10,11,12,13]. According to Yoon et al. [9], AAV is smaller than RV and AD, which facilitates the penetration of the ZP during co-cultivation.

Takahashi et al. [15] and Ohtsuka et al. [16] developed novel methods (GONAD and *i*-GONAD) to induce genome editing at the target loci in situ that do not require ex vivo handling of embryos. Unfortunately, this technology requires an expensive electroporator (the main EP apparatus). For convenient and easy manipulation of early embryos, it would be ideal to employ substances that allow for direct gene delivery through the ZP using GONAD/*i*-GONAD-based technology. To date, at least three ZP-penetrating substances (chitosan, multiwall carbon nanotubes [MWNTs], and ViewFect) have been developed [29,30,31], although no successful genome editing attempts have been reported using these substances. AAV appears to be the only agent that enables genome editing (knock out and knock-in) of ZP-intact preimplantation embryos in vivo and in vitro [8,9,10,11,12].

Highly concentrated AAV vectors can achieve high rates of transduction in preimplantation embryos. According to Yoon et al. [9], zygotes treated with 6 × 10^8^ GCs in 10 μL of an explant medium (i.e., 6 × 10^10^ GC/mL) for one day resulted in a 100% indel frequency after ET to recipient females. However, the editing frequency dropped to 25% for Day 16.5 embryos and 20% for newborns at 6 × 10^7^ GCs. No edited animals were detected in a 6 × 10^6^ GC treatment group. In the present study, no fluorescence was observed in the morulae or oviductal epithelium after 1 μL injection of rAAV-CAG-EGFP-6 at a concentration of 4 × 10^11^ GC/mL (data not shown), which was approximately a tenth of the concentration used in the main experiments. Thus, gene delivery, as well as gene editing efficiency, is rAAV dose dependent.

Similar to Yoon et al. [9], we observed that AAV-mediated infection of early embryos was possible in vivo when scAAV-CAG-EGFP-6 was injected into the oviduct of pregnant female mice at Day 0.7 of pregnancy (at the late zygote stage). In vivo EP of entire oviducts after intra-oviductal instillation of viruses could enhance the uptake of viral particles by preimplantation embryos through GONAD/*i*-GONAD using RNP or *Cas9* mRNA/gRNA [15,16]. However, this was not apparent in the present study where scAAV-CAG-EGFP-6 (3 × 10^12^ GC/mL) was subjected to intra-oviductal instillation and subsequently to in vivo EP (data not shown). Unlike the electrical capacity of naked nucleic acids, AAV particles may not have the same potential, because their nucleic acids are tightly packaged within a capsid.

To effectively generate GM animals, it is important to know when, and for how long, AAV infection occurs in early mouse embryos using rAAVs. In Yoon et al. [9], zygotes were transduced overnight with rAAV-6 carrying CRISPR/Cas9 components and the resulting 2-cell embryos were then transferred into pseudo-pregnant recipients. Zygotes treated with 6 × 10^8^ GCs of rAAVs resulted in a 100% indel frequency, suggesting that 1-day-incubation in the presence of rAAV is sufficient for AAV-mediated gene modification in early mouse embryos. In the present study, we observed EGFP fluorescence in 2-cell embryos recovered one day after AAV-based GONAD, although the GFP intensity was weaker than that observed at the morula stage (see Figure 3A(a–c) vs. Figure 3A(d–f)). To achieve maximal genome editing performance, it would be ideal to treat embryos with a high concentration of rAAVs for at least two days, up to the morula stage of development.

Mosaicism is a major complication of gene editing in early mouse embryos. This is especially induced at the later stages of embryo development (e.g., 2-cell embryos) (reviewed by Sato et al. [17]). In AAV vectors, gene expression-related slow onset often occurs, probably due to the time-consuming conversion of single-stranded to double-stranded AAV genomes (reviewed by McCarty [32]). Romeo et al. [13] demonstrated that AAV expression commenced from morulae and continued up to blastocysts when zygotes were continuously incubated with rAAV-1 (or DJ) for 4 d. Similar to Romeo et al. [13], EGFP expression peaked at the morula stage when zygotes were treated with scAAV-CAG-EGFP-6 in vivo for 1 d and allowed to cultivate in vitro until the late blastocyst stage (see Figure 3B). No apparent mosaicism was observed in our experiments (Figure 1A and Figure 3B).

Toxicity towards early embryos is another issue associated with AAV-mediated gene delivery. This phenomenon is evident in mouse zygotes infected in vitro by human helper-dependent AAV-2 [33] or nonhuman primate embryos infected with rAAV-6 [14]. However, the level of toxicity depends on the AAV serotype used. Romeo et al. [13] demonstrated that rAAV-DJ (conferring EGFP expression under the elongation factor 1α promoter) was more deleterious than rAAV-1, although both rAAVs infected early mouse embryos (from morulae to blastocysts) with high efficiencies. For scAAV-CAG-EGFP-6, no appreciable level of toxicity towards mouse preimplantation embryos was observed (Figure 3B(a,b,e,f,i,j,m,n)), similar to the findings of previous reports [8,9]. Thus, rAAV-6 (and possibly rAAV-1) is a safe viral vector that allows efficient delivery of exogenous genes of interest across the ZP of mammalian embryos.

The oviduct facilitates sperm transport, fertilization, ovum and embryo transport, and embryonic development. The secretory cells in the epithelia of mammalian oviducts produce and release various agents into the lumen that maintain a conducive environment for early embryo development [34,35]. Several oviduct-specific genes have been identified and their functions investigated [36,37,38]. However, much remains unknown about these oviduct-specific proteins and the function of the oviduct itself. This is apparent in the general lack of efficient gene delivery systems for these cells, excluding EP or liposome-based gene delivery approaches [18,39,40]. According to our results, AAV-based GONAD enabled efficient transduction of oviductal epithelial cells and could be a convenient tool for genome manipulation in these cells.

Along with other modern advances in genome editing, especially CRISPR-based platforms, AAV vectors are potential tools for correcting gene mutations in genetic disorders [41]. Gene knock-in has been performed through co-transduction with rAAV carrying a *Cas9* and gRNA expression unit [9], sequential transduction (in vitro EP in the presence of RNP and donor DNA) and subsequent transduction with rAAV carrying *Cas9* [8,12] or transduction with an all-in-one AAV vector carrying *Neisseria meningitidis* Cas9 (*NmeCas9*) and gRNA [42]. Furthermore, infection with a rAAV coupled CRISPR/Cas9 system enabled targeted insertion of large fragments [43]. The present study could prove useful when AAV-based GONAD is coupled with an established gene-editing system.

## 4. Conclusions

The present study evaluated the ability of scAAV vectors to transduce ZP-enclosed zygotes in situ by injecting separate solutions containing several serotypes of scAAV (carrying a reporter gene expression unit) into the oviductal lumen of pregnant female micee. ScAAV serotype 6 showed very high rates of transduction (Figure 1B(a–c), Figure 2A(i–k) and Figure 3B ), although the fluorescence intensity varied among embryos. The reporter gene expression (as evaluated by green fluorescence) was discernible at the 2-cell stage, one day after the AAV-based GONAD, and peaked at the morula stage. Most of the embryos lost their fluorescence at the late blastocyst stage. This transient nature of the AAV-based gene transfer system could benefit studies of gene function in early-stage embryos and improve productivity in the generation of genome-edited animals. Furthermore, oviductal epithelial cells facing oviductal lumen were efficiently transduced, suggesting that the AAV-based GONAD could be a useful tool in manipulating the genomes of these cells. The numbers of animals and the titers of rAAV vectors are summarized in Table 1. The AAV-based GONAD does not require special equipment, such as manipulators and electroporators, and facilitates convenient and reliable genome manipulation in the female mouse reproductive system.

## 5. Materials and Methods

### 5.1. Mouse and Superovulation Induction

Adult (4- to 6-month-old) female B6C3F1 mice (a C57BL/6 and C3H/He hybrid, Kyudo, Tosu, Japan) were used in the experiment. The mice were maintained in a 12 h light/dark cycle (lights on from 07:00 h to 19:00 h) and were provided with food and water ad libitum.

For superovulation induction, the females were intraperitoneally (IP) injected with 5 IU of pregnant mare serum gonadotropin (PMSG; eCG) at approximately 11:00 h, followed by administration of the same dose of human chorionic gonadotrophin (hCG) 48 h later [44]. Immediately after the injection of hCG, the females were allowed to mate with fertile males. The following morning, successful copulation was determined by the presence of a vaginal plug, which was considered to be Day 0 of pregnancy. Only females with vaginal plugs were subjected to AAV-based GONAD treatment on Day 0.7 of pregnancy (approximately 11:00 h), which corresponds to the late zygote (1-cell embryo) stage.

### 5.2. RAAV Vectors

The plasmid used to produce green fluorescent protein (GFP)-expressing scAAV was obtained from Addgene (#83279, pscAAV-CAG-EGPF), which contained an *EGFP* cDNA sequence under the CAG [45], a promoter comprised of a cytomegalovirus (CMV) enhancer and chicken β-actin promoter. The packaging plasmids encoding for a capsid protein (serotype 1, 5, or 6) and a replicase were purchased from Cell Biolabs (#VPK-421, pAAV-RC1, San Diego, CA, USA) or Addgene (#219845, pAAV2/2; #201372, pAAV2/5; #212291, pDGM6). The helper plasmid (#340202, pHelper) for AAV production packaging was obtained from Cell Biolabs. The resultant rAAVs were termed scAAV-CAG-EGFP-1, -2, -5, and -6, which correspond to rAAV serotype 1, 2, 5, and 6, respectively.

### 5.3. RAAV Vector Production and Purification

RAAVs were generated by polyethylenimine-based transfection of the plasmid cocktail (pscAAV-CAG-EGFP, pHelper, and a packaging plasmid) into HEK293T cells (#RCB2202, RIKEN BioResource Center, Tsukuba, Japan). Four days after transfection, cells were harvested for purification of rAAVs. Briefly, the transfected HEK293T cells were lysed and viral particles were extracted using a purification kit (#6235, AAVpro extraction solution, Takara Bio, Kusatsu, Japan). Viral particles were also recovered as pellets from the culture medium by centrifugation at 7500× *g* for 12 h at 6 °C. Crude lysates and viral pellets were suspended in phosphate-buffered saline (PBS) and treated with nuclease (#2670, Cryonase cold-active nuclease, Takara Bio). The supernatant was then affinity-purified with anti-AAV capsid antibody-coated beads (#A36740, POROS CaptureSelect AAVX resin, Thermo Fisher Scientific, Waltham, MA, USA, or #28411201, ABV Sepharose, Cytiva, Marlborough, MA, USA). Purified AAV vectors were concentrated by ultrafiltration using an Amicon ultra filter (#UFC910024, Ultracel-100K, Merck Millipore, Burlington, MA, USA). The titers of the AAV vector stocks were determined by real-time PCR using a SYBR reaction mixture (#RR420A, SYBR premix Ex Taq, Takara Bio) and a primer set for the woodchuck hepatitis virus posttranscriptional regulatory element (WPRE) sequences (WPRE-F1, CCCAAAGGGAGATCCGACTCGT; WPRE-R1, AATCCAGCGGACCTTCCTTCCC).

### 5.4. Transduction of Cells with rAAVs

HEK293T cells were grown in Dulbecco’s modified Eagle’s medium (DMEM) containing 10% fetal bovine serum (FBS) and penicillin/streptomycin. Viral vector stocks were diluted (1–2 × 10^9^ GC/mL of medium), added to the sub-confluent cultures, and left for 3 d. Green fluorescence, derived from transduced EGFP-expressing rAAVs in HEK293T cells, was recorded under a fluorescent microscope (IX70, Olympus, Tokyo, Japan) with a CMOS camera (Wraymer, Osaka, Japan).

### 5.5. AAV-Based GONAD

The procedure of AAV-based GONAD is essentially the same as *i*-GONAD [16], except for using in vivo EP. Briefly, adult B6C3F1 female mice at Day 0.7 of pregnancy (corresponding to the late 1-cell zygote stage; at approximately 16:00 h) were anesthetized by intraperitoneal injection of three combined anesthetics (medetomidine, midazolam, and butorphanol) [46]. Then, a small incision was made in the central dorsal skin of the anesthetized mouse, and the organs (ovary, oviduct, and uterus) associated with adipose tissue were exposed. 

The oviductal injection solution was made by mixing 4 μL of rAAV (2–6 × 10^12^ GC/mL) and 1 μL of 0.4% Fast Green FCF (#15939-54, Nacalai Tesque, Kyoto, Japan; used to enable visual identification of the injected solution). A glass needle (tip diameter of 15–30 μm, connected to a mouthpiece, a cotton plugged 200 μL-tip) was used to collect 1 μL of the solution, which was injected via the oviductal wall just in front of the ampulla with the aid of a dissecting microscope (SZX10, Olympus). The treated oviduct was returned to its original position and the same treatment was performed on the opposite oviduct. After the operation, the wound was closed with wound clips. The anesthetized mice were recovered with a subcutaneous injection of atipamezole, a medetomidine antagonist [46]. A second group of untreated mice served as a control.

### 5.6. Embryo Collection and Oviduct Processing

Morula stage embryos (corresponding to Day 2.7 of pregnancy, including 8-cell embryos) were collected from the oviducts of mice subjected to AAV-based GONAD by flushing the oviducts with Dulbecco’s modified Ca^2+^, Mg^2+^-free phosphate-buffered saline (DPBS) containing 0.3% FBS (hereafter referred to as DPBS-FBS). The embryos were then subjected to fluorescence analysis under an inverted microscope (#DMI4000, Keyence, Itasca, IL, USA) with epifluorescence. In some cases, 2-cell stage embryos (corresponding to Day 1.7 of pregnancy) were collected from the oviducts by flushing the oviducts with DPBS-FBS. The collected embryos were then transferred to mWM medium covered with paraffin oil (ARK Resource, Kumamoto, Japan) in a Terasaki microtest plate (Nunc, Roskilde, Denmark) for cultivation at 37 °C in 5% CO_2_ until the late blastocyst stage. EGFP-derived green fluorescence was periodically observed at the 2-cell (immediately after culture), morula (1 day after culture), early blastocyst (2 days after culture), and late blastocyst (3 days after culture) stages.

The oviducts from which 2-cell embryos were collected were placed between a coverglass and a culture dish to observe the expression of green fluorescence caused by rAAV infection in the oviductal epithelium (facing oviductal lumen).

### 5.7. Observation of Fluorescence and Quantitative Evaluation

Bright-field and fluorescence images of mouse embryos (2-cell embryos, morulae, early blastocysts, or late blastocysts) and oviducts were captured using an inverted microscope (BZ-X810, Keyence). Structured illumination was used to obtain optically sectioned z-stack images (Appendix A).

The average fluorescence intensity of each embryo was measured using Image J software (Fiji version) [47]. Raw images were processed using rolling ball background subtraction. ROIs were manually set. Fluorescence data were visualized and statistically analyzed using Graphpad Prism software version 7/8.

## Figures and Tables

**Figure 1 ijms-23-04897-f001:**
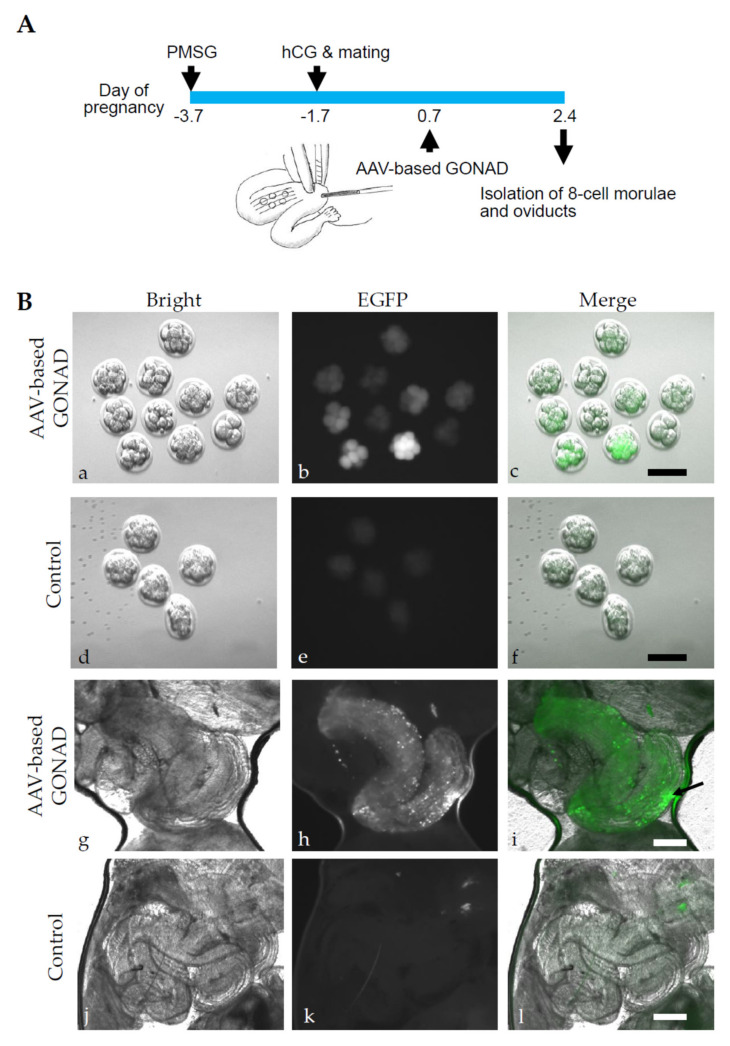
Isolation of morulae and oviducts 2 d after AAV-based GONAD in pregnant B6C3F1 females using scAAV-CAG-EGFP-6. (**A**). Experimental scheme. The B6C3F1 females first received a hormonal treatment of pregnant mare serum gonadotropin (PMSG) and human chorionic gonadotrophin (hCG) to induce superovulation. The morning after mating with males, females were inspected for vaginal plugs. The pregnant females (those with vaginal plugs) were next subjected to AAV-based GONAD at Day 0.7 of pregnancy (corresponding to the late zygote stage). On Day 2.4 of pregnancy, morulae and oviducts were isolated for EGFP-derived fluorescence analysis. (**B**). Inspection of fluorescence in morulae (**a**–**f**) and oviducts (**g**–**l**). (**a**–**c**) Morulae from the females subjected to AAV-based GONAD. (**d**–**f**) Morulae from the untreated females (used as controls). (**g**–**i**) Oviduct of a female subjected to AAV-based GONAD. Note that fluorescent area corresponds to the oviduct (ampulla) injected with rAAV vcetor. (**j**–**l**) Oviduct of an untreated female. Bright, photographs taken under white light; EGFP, photographs taken under blue light illumination; Merge, mixed images of white and blue light illumination. Scale bars, 100 μm (**a**–**f**) and 500 μm (**g**–**l**).

**Figure 2 ijms-23-04897-f002:**
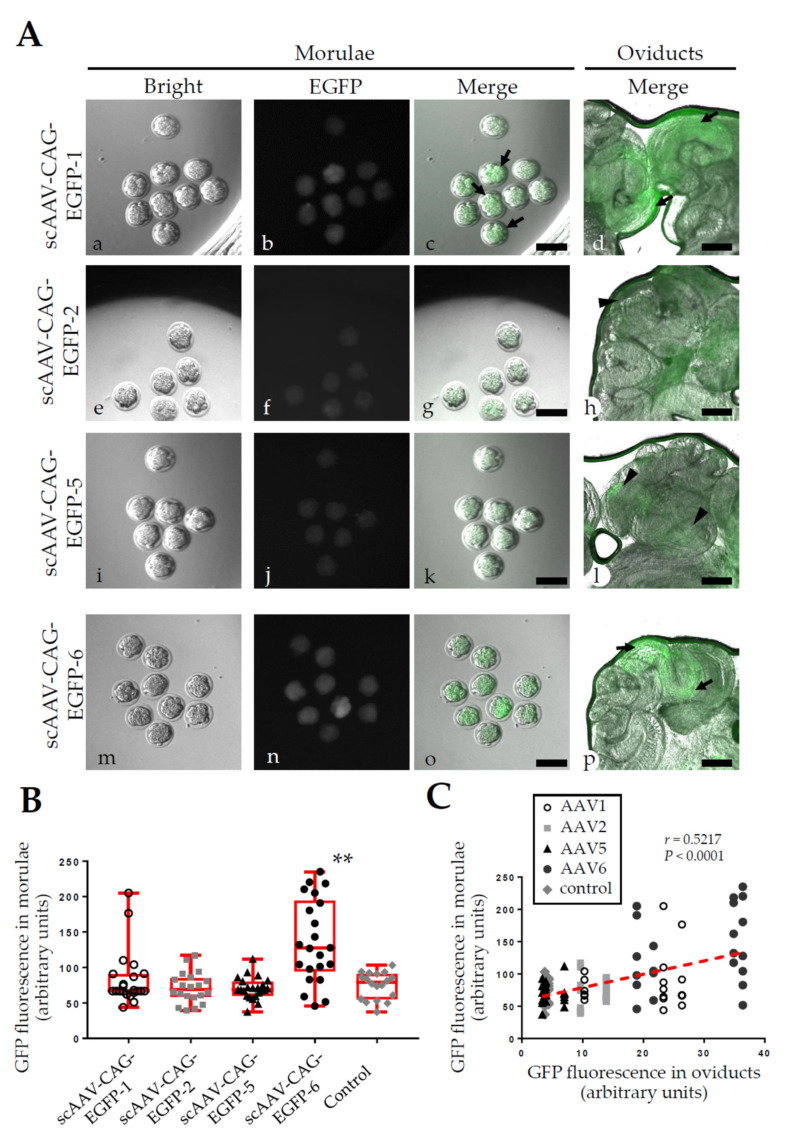
Tropism of rAAV-1, -2, -5, and -6 for mouse embryos and oviducts. (**A**). Monitoring of fluorescence in morulae (**a**–**c**,**e**–**g**,**i**–**k**,**m**–**o**) and oviducts (**d**,**h**,**l**,**p**) 2 days after AAV-based GONAD in pregnant B6C3F1 females using scAAV-CAG-EGFP-1, -2, -5, and -6. Bright, photographs taken under white light; EGFP, photographs taken under blue light illumination; Merge, mixed images of white and blue light illumination. Scale bars, 100 μm (**a**–**c**,**e**–**g**,**i**–**k**,**m**–**o**) and 500 μm (**d**,**h**,**l**,**p**). (**B**). Quantification of GFP expression in individual morulae treated with different serotypes of the rAAV vectors. The box and whisker plots with maximum and minimal ranges correspond to all data points analyzed. A Kruskal–Wallis test with Dunn’s multiple comparisons revealed a significant difference between the control groups and the group transduced with scAAV-CAG-EGFP-6 (** *p* < 0.005). (**C**). Correlation analysis on mean GFP expression levels in individual morulae and those in the oviducts from which the morulae were isolated. A modest correlation was found between transduction in morulae and that in oviductal epithelial cells (Pearson’s correlation coefficient *r* = 0.5217, *p* < 0.0001). A fitting line to all data points is shown in red.

**Figure 3 ijms-23-04897-f003:**
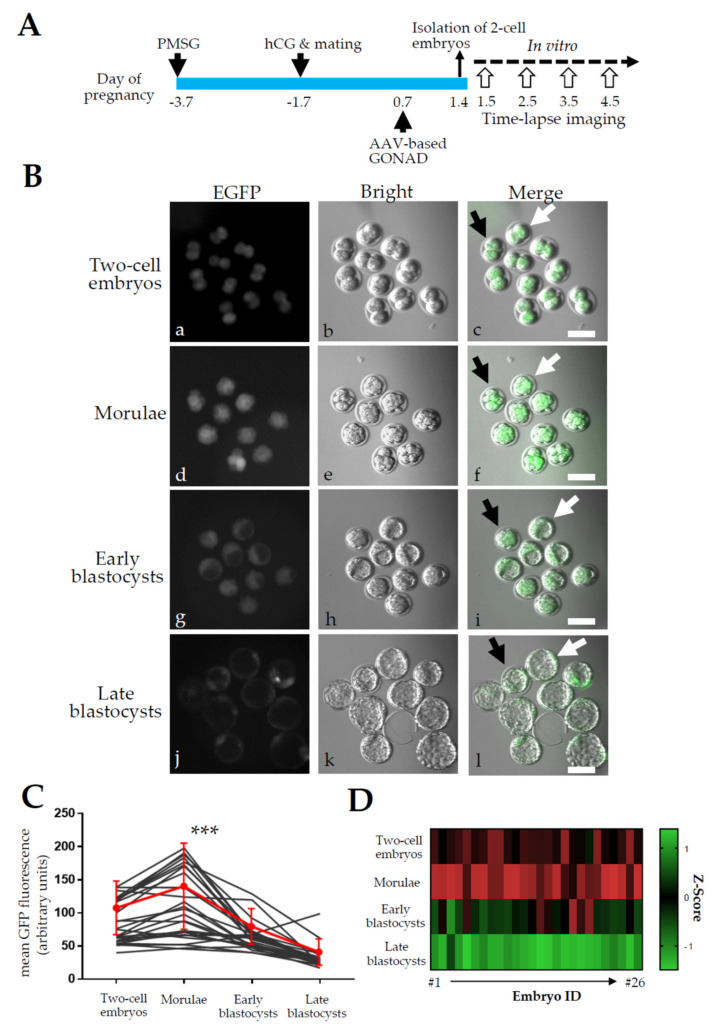
Time-lapse imaging of fluorescence expression in mouse embryos after AAV-based GONAD using scAAV-CAG-EGFP-6. (**A**). Experimental scheme. The B6C3F1 females first received a hormonal treatment of PMSG and hCG to induce superovulation. The morning after mating with males, the females were inspected for vaginal plugs. Females were next subjected to AAV-based GONAD at Day 0.7 of pregnancy (corresponding to the late zygote stage). On Day 1.4 of pregnancy, 2-cell embryos were isolated and cultured until the late blastocyst stage to assess EGFP-derived fluorescence over time. Imaging was repeatedly carried out on the same embryos everyday from Day 1.5 to Day 4.5. (**B**). Time-lapse imaging of fluorescence in 2-cell embryos (**a**–**c**), morulae (**d**–**f**), early blastocysts (**g**–**i**), and late blastocysts (**j**–**l**). Black and white arrows indicate the same embryos over time. Scale bar, 100 μm. (**C**). Black lines indicate individual embryos and the red line represents mean values with error bars showing the standard deviation. Repeated measures 1-way ANOVA with Tukey’s multiple comparison reveals a significant main effect of the developmental stage (F_(2, 75)_ = 50.25, *p* < 0.0001) and the highest GFP expression at the morula stage (***, *p* < 0.0003). (**D**). A heat-map showing normalized time-course changes in fluorescence expression for individual embryos. The same data shown in (**C**) were transformed to z-scores for each embryo, and expressed as a heat-map.

**Table 1 ijms-23-04897-t001:** Numbers of animals, embryos and viral titers used in this study.

Experiment	AAV Titer (GC/mL)	Number of Animals Examined	Number of Oviducts Examined	Number of Embryos Examined
**Experiments 1 & 2**				
scAAV-CAG-EGFP-1	2.1 × 10^12^	2	3	22
scAAV-CAG-EGFP-2	3.4 × 10^12^	2	2	20
scAAV-CAG-EGFP-5	1.9 × 10^12^	3	3	23
scAAV-CAG-EGFP-6	3.4 × 10^12^	4	5	32
Control (uninfected)	n/a	3	3	23
**Experiments 3**				
scAAV-CAG-EGFP-6	6.0 × 10^12^	3	3	26

n/a, not applicable.

## Data Availability

All data are available from the authors upon reasonable request.

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
