# Peer review of "Direct Injection of Recombinant AAV-Containing Solution into the Oviductal Lumen of Pregnant Mice Caused In Situ Infection of Both Preimplantation Embryos and Oviductal Epithelium"

_ijms, 2022, doi:10.3390/ijms23094897_

Round 1
Reviewer 1 Report
Summary
Sato et al. have developed a method to introduce transgenes into pre-implantation embryos in the zona pellucida using AAV6. This is likely to be a useful tool in the generation of genetically modified animal models. Using EGFP as a reporter, the authors screen multiple AAV serotypes (AAV1, AAV5 and AAV6) to identify AAV6 as the most effective for in vivo transduction. The authors also determine which stage of pre-implantation embryos are most effective for in vivo transduction. Finally, they determine that transgene expression can be detected at the two cell stage and peaked at the morula stage. By building on these principles, future investigations can express CRISPR reagents to generate novel genetically modified animals. Therefore, these findings are novel development in the field of gene therapy and animal modelling; and will facilitate more sophisticated studies in the future. Pending some minor comments, the manuscript is suitable for publication in the International Journal of Molecular Sciences.
Comments
- Please provide scale bars in Figure 1 panels c and i. These are distinct from the control samples and should be treated as such.
- Figure 1 provides compelling representative images. It would be nice to provide statistics and measurements, or at the very least a description of the number of animals used in this study.
- Fig 3C- error bars on either side of the mean will be helpful
- Line 356 – please provide primer sequences for the primers used to titer the vectors or cite the appropriate reference.
Author Response
Comments and Suggestions for Authors
Summary
Sato et al. have developed a method to introduce transgenes into pre-implantation embryos in the zona pellucida using AAV6. This is likely to be a useful tool in the generation of genetically modified animal models. Using EGFP as a reporter, the authors screen multiple AAV serotypes (AAV1, AAV5 and AAV6) to identify AAV6 as the most effective for in vivo transduction. The authors also determine which stage of pre-implantation embryos are most effective for in vivo transduction. Finally, they determine that transgene expression can be detected at the two cell stage and peaked at the morula stage. By building on these principles, future investigations can express CRISPR reagents to generate novel genetically modified animals. Therefore, these findings are novel development in the field of gene therapy and animal modelling; and will facilitate more sophisticated studies in the future. Pending some minor comments, the manuscript is suitable for publication in the International Journal of Molecular Sciences.
We deeply appreciate all comments from the reviewer. These comments improve our manuscript and also much encourage us to explore our future study using this AAV-based GONAD system.
Comments
- Please provide scale bars in Figure 1 panels c and i. These are distinct from the control samples and should be treated as such.
Answer: We agree with you. As suggested, we placed scale bars in Figure 1-c, i of the revised Figure 1. We have also attached scale bars in Figure 2 g, k, o, h, l, p as well as Figure 3 f, i, j, accordingly.
- Figure 1 provides compelling representative images. It would be nice to provide statistics and measurements, or at the very least a description of the number of animals used in this study.
Answer: We are glad to hear that the reviewer appreciates the representative images in Figure 1. We agree that some stats for animals would be necessary. Therefore, we have newly included Table 1 to indicate the numbers of animals, ovaries/oviducts and embryos as well as the titers of AAV vectors. We have also included quantification of GFP signals using optical sectioning images for AAV-based GONAD embryos in the supplemental figure (Figure S2), which was requested by the other reviewer. Furthermore, in order to provide better contrasts of GFP signals and background, we have replaced all green color images into grayscale images in the main figures. We believe that these changes would facilitate better understanding of our data.
- Fig 3C- error bars on either side of the mean will be helpful
Answer: We thank this suggestion. As suggested, we redrew the previous Figure 3C, where error bars (SD) are shown in the revised text.
- Line 356 – please provide primer sequences for the primers used to titer the vectors or cite the appropriate reference.
Answer: We apologize that this important information was missing in the previous manuscript. Now, we have added the primer sequence information used for quantification of viral titers in the revised text (L386 to L388).
Reviewer 2 Report
Sato et al describe a protocol to transduce pre-implantation embryos and oviductal epithelium with recombinant Adeno-Associated Viral vectors bearing the Green Fluorescent Protein as a reporter gene. Adeno-associated vectors are widely used to transduce different tissues and their application in gene-editing is well described by hundreds of published papers.
Provided results in support to the protocol described by authors are not sufficient to understand the novelty of their approach.
Here my concerns
Figure 1 provides evidence of transduction of morulae and oviducts, while it seems clear that AAV-based GONAD protocol efficiently transduced oviducts, it is not clear what is the efficiency of morulae transduction, considering that controls also scored positive for GFP, maybe confocal microscopy can help to fix this issue. How many morulae were analyzed? To calculate the efficiency of transduction, authors have to provide better statistics. From the images provided, it seems that just 3/10 morulae were transduced.
Result 2 claims a strong positive correlation between transduction performance in the embyos and oviductal epithelial cells but this positive correlation must be measured with statistics. Moreover, the graphs in figure 2 reported relative GFP fluorescence but it is not clear relative to what. To the controls? Are the values a % or an index? If it is not a % or a fraction or a fold increase please fix the text and provide a unit.
Also the third experiment is a bit confusing. Authors want to check the duration of EGFP expression but the provided illustration did not clarify how long the authors have checked for GFP production. Also in this case, a relative quantification of AAV genomes in the tissues/cells might provide a better method to their question: how long do the vector last. Also in figure 3C it is not clear how the statistics were made. It appears to me that strong variability created several analytical issues. This might be solved, again, with qPCR performed at different time-points
Minor revision
Cas9 is not Caspase9!!!
Authors erroneously use transduction and infection as synonyms but they are not. Also the title is mis-leading. They are using adeno-associated viral vectors, not Adeno Associated Viruses.
Author Response
Comments and Suggestions for Authors
Sato et al describe a protocol to transduce pre-implantation embryos and oviductal epithelium with recombinant Adeno-Associated Viral vectors bearing the Green Fluorescent Protein as a reporter gene. Adeno-associated vectors are widely used to transduce different tissues and their application in gene-editing is well described by hundreds of published papers. Provided results in support to the protocol described by authors are not sufficient to understand the novelty of their approach.
Here my concerns
- Figure 1 provides evidence of transduction of morulae and oviducts, while it seems clear that AAV-based GONAD protocol efficiently transduced oviducts, it is not clear what is the efficiency of morulae transduction, considering that controls also scored positive for GFP, maybe confocal microscopy can help to fix this issue. How many morulae were analyzed? To calculate the efficiency of transduction, authors have to provide better statistics. From the images provided, it seems that just 3/10 morulae were transduced.
Answer: We appreciate the comments and understand the reviewer’s concern. As pointed out by the reviewer, we admit that there is variability in AAV-based transduction efficiency, depending on various factors including the serotypes of AAV vectors as well as infection efficiency. The infection efficiency highly depends on the concentration of viral vectors in the oviducatal lumen and the location of embryos in the oviduct in vivo, both of which are difficult to control precisely. Furthermore, the GFP expression levels used as an index of gene transduction in this study are likely depending on how many viruses are incorporated per cell/embryo. Due to the variability and possible graded expression of GFP, in this study, we have decided not to explicitly classify AAV vector-treated embryos into GFP positive and negative, and thus not to calculate the transduction efficiency. Instead, we just indicate “relative” GFP fluorescence levels in Figures 2 and 3 in this study. (We apologize the confusion of the term “relative”, which we will explain below.) However, we totally agree with the reviewer that confocal microscopic images would help to distinguish GFP signals from autofluorescence/background signals. In fact, we have analyzed GFP signals using optical sectioning images of embryos shown in Figure 1B. We used a structured illumination-based optical sectioning system instead of confocal laser scanning systems, just because of the availability in our animal facility. As shown in the newly added Supplemental Figure 2 (Figure S2), the optical sectioning analysis nicely demonstrated that many of the AAV-treated embryos give GFP signals even in a single z-plan image (Figure S2-A and -B). We have measured the mean GFP fluorescence signals for each embryo and the measures were shown in Figure 2S-C. We assume that 3 out of 10 morulae are GFP negative (indicated by arrows in Figure S2-A) in this case. We have included the above discussion in the revised manuscript (L138-L144). Concerning the question about “How many morulae were analyzed? or authors have to provide better statistics”, we have newly added Table 1 in order to provide information about the numbers of animals and embryos used in the revised manuscript. The statistical analysis on differences in GFP expression levels across AAV serotypes were shown in Figure 2B of the previous manuscript. In the revised manuscript, we have now included the data of serotype 2 for better understanding of viral tropism (please see Figure 2B in the revised text). We have revised the text accordingly (L179-L193). Finally, in order to provide better contrasts of GFP signals and background for readers, we have replaced all green color images into grayscale images in the main figures (please see Figure 1B-b,e,h,k, Figure 2A-b,f,j,n and Figure 3B-b,e,h,k in the revised text). We believe that these changes would facilitate better understanding of our data.
- Result 2 claims a strong positive correlation between transduction performance in the embryos and oviductal epithelial cells but this positive correlation must be measured with statistics. Moreover, the graphs in figure 2 reported relative GFP fluorescence but it is not clear relative to what. To the controls? Are the values a % or an index? If it is not a % or a fraction or a fold increase please fix the text and provide a unit.
Answer: We deeply thank the reviewer for raising these critical comments. In the previous text, we mentioned that there is a strong positive correlation between transduction performance in the embryos and oviductal epithelial cells (L182-L183 in the previous manuscript). This statement was based on our correlation analysis of GFP signals in each morula and in the oviduct from which morulae were isolated. In the revised manuscript, we have included the data from this analysis in Figure 2C. This panel indicates a statistically significant correlation between GFP signals between embryos and oviducts (P < 0.0001). However, during the course of this analysis, we have noticed that the correlation coefficient is not very high (r = 0.5217). Thus, we have de-emphasized this sentence in the revised manuscript, accordingly (L194-196).
As to the “relative” GFP fluorescence, we feel sorry for causing any confusion which the reviewer has had. The term “relative” is a jargon in our field, meaning that GFP fluorescence could not be expressed in an “absolute” unit because fluorescence values change even though the same sample is imaged, depending on various parameters such as exposure time, intensity of excitation light, choice of filters, and light path, etc. Thus, we often used “relative” fluorescence, but it doesn’t mean any percentage or fold increase to a control or to something else. In response to the reviewer’s concern, in the revised manuscript, we have now rephrased the index labeling in all figures. We now use the term “GFP fluorescent (arbitrary units)”, meaning the signal levels expressed arbitrarily. Not to mention, the imaging conditions/parameters were fixed in the same experiments for the same tissues/targets so that the values can be comparable/evaluatable each other.
- Also the third experiment is a bit confusing. Authors want to check the duration of EGFP expression but the provided illustration did not clarify how long the authors have checked for GFP production. Also in this case, a relative quantification of AAV genomes in the tissues/cells might provide a better method to their question: how long do the vector last. Also in figure 3C it is not clear how the statistics were made. It appears to me that strong variability created several analytical issues. This might be solved, again, with qPCR performed at different time-points.
Answer: We understand the concern and confusion raised by the reviewer. The reviewer correctly pointed out that, in the 3rd experiment, we aimed to know how long expression of EGFP fluorescence lasts in the early embryos after AAV-based GONAD. As the reviewer suggested, quantitation of rAAV-derived transgene through qPCR in embryos after AAV-based GONAD would be one of the approaches to this issue. However, as discussed above, there is variability of AAV infection and/or transgene transduction in individual embryos, for which single cell-based repeated measures analysis would be ideal to obtain time course data. Unfortunately, the current technology does not allow us to carry out qPCR measurement without cell lysis, thus impossible to perform repeated measures for the same embryos over time. Therefore, we employed time-lapse imaging to quantify GFP signals from individual embryos over four days (from Day 1.5 to Day 4.5 of pregnancy). Unfortunately, as the reviewers pointed out, we missed to explicitly indicate the duration of our time-lapse imaging. This important information has been included in both Figure 3A and the text (L215-L216). We deeply thank the reviewer for arising this important issue. It is well recognized that the AAV genome is generally unable to be integrated into host chromosomes and gradually lost during cell division. Along with this, fluorescence generated from rAAV carrying GFP gene disappears. This means a close relationship between the amounts of AAV present in a cell and fluorescence intensity generating from EGFP expression. Thus, it is easily expected that AAV introduced at early stage of embryogenesis will disappear in later stage of development (at least up to blastocyst stage), because zygotes exhibit extensive cell division called blastomere cleavage. Our final purpose of this study is to grasp how long expression of fluorescence lasts during early embryogenesis. In this context, that expression of EGFP from the introduced rAAV reach at morula stage is informative for us, because AAV-based GONAD can be efficiently done at least up to morula stage (Day 2.5 of gestation). As to statistical tests for Figure 3C、we unintentionally failed to describe our 1-way ANOVA with repeated measures. In the revised manuscript, we have now added our ANOVA data to describe significant differences across embryo stages (L218-L221). Even though there are embryo-to-embryo differences in fluorescent signals, ANOVA detects statistically significant differences across embryonic stages (P < 0.0001). Post-hoc multiple comparison further indicates that the morula stage exhibited highest expression of GFP after AAV-based GONAD, which was different from any other stages (Ps < 0.0003).
Minor revision
- Cas9 is not Caspase9!!!
Answer: We deeply thank the reviewer for pointing out this. This is quite embarrassing for us as molecular biologists. It must have happened during English editing service by who may not be very familiar with molecular biology. But it was purely our fault. Again, thank you.
- Authors erroneously use transduction and infection as synonyms but they are not. Also the title is mis-leading. They are using adeno-associated viral vectors, not Adeno Associated Viruses.
Answer: We appreciate these comments raised by the reviewer. We agree that gene transduction and viral infection are not the same. We have carefully checked through the text and have corrected where the usage was inappropriate. We also agree with the reviewer that AAVs and AAV vectors are different. However, as to our title, we describe “Recombinant AAV-Containing Solution”. Because in many studies and literatures, the term “recombinant AAV” has been used as the same meaning for “AAV vector”, and because what we actually did was “injection of recombinant AAV-containing solution into the oviductal lumen”, we are wondering the possibility of misleading in our current title. We would be happy to revise our title if the reviewer kindly explains more about what causes misleading/confusion.